# A Strategy for Sample Preparation: Using Egg White Gel to Promote the Determination of Aflatoxin M1 Content in Milk Samples

**DOI:** 10.3390/molecules27155039

**Published:** 2022-08-08

**Authors:** Xiao Ning, Lulu Wang, Shaoming Jin, Xuran Fu, Xiulan Sun, Jin Cao

**Affiliations:** 1School of Food Science and Technology, Jiangnan University, Wuxi 214122, China; 2Key Laboratory of Food Quality and Safety for State Market Regulation, National Institute of Food and Drug Control, Beijing 100050, China

**Keywords:** egg white gel, matrix-purifying material, aflatoxin M1, milk, green analytical chemistry

## Abstract

The analysis of food samples is a challenging task. The high complexity of food matrices hinders the extraction and detection of analytes from them. Therefore, the correct preparation of food samples is a crucial step for their subsequent analysis, as it achieves the proper isolation and preconcentration of analytes and removes the interfering proportion of the food matrix before instrumental analysis. We aimed to develop a method that not only satisfies the requirement of detecting trace compounds in complex matrices but also achieves a “greener” approach by reducing the use of organic solvents and non-degradable materials to minimize the health hazards posed to the operators as well as pollution to the environment. In this study, we prepared egg white as a concentrated gel and used this material for the biological purification of milk samples. After the milk protein was removed by acidification and salting, the residual amount of aflatoxin M1 in milk samples was quantitatively determined by ultra-performance liquid chromatography-tandem mass spectrometry (UPLC-MS/MS). The results showed that the novel egg white purification method possessed advantages over the immunoaffinity technique used as the reference method in extraction recovery, sensitivity, repeatability, and operability. The limit of detection (LOD) was 0.001 μg/kg. In spiked samples containing 0.01 μg/kg to 2 μg/kg of AFM1, the average recovery was 88.3–94.7%, with a precision of 6.1–11.0%. Improved repeatability was obtained by significantly reducing the operation time and resource requirements compared with the immunoaffinity technique currently used internationally. This study provides a reference for the further improvement of the relevant international standards in place for the detection of aflatoxin M1 in milk.

## 1. Introduction

In conducting experiments, components are often enriched for their subsequent measurement while excluding other interfering substances that would otherwise undermine the results’ reliability or validity [1]. Therefore, the proper processing of samples is key to food nutrition and safety testing and in the context of any research that utilizes food components [2]. Existing studies on sample clean-up and enrichment technologies have mainly focused on studying specific, highly-purified materials. In recent years, with the gradually improved collective understanding of “green analytical chemistry” [3,4,5,6], there is a new focus on the methods employed to achieve extraction, purification, and matrix replacement with natural substances [7].

As a toxic substance, the World Health Organization classified aflatoxin B1 (AFB1) as a class 1 carcinogen [8]. Evidence has shown that AFB1 causes damage to both human and animal liver tissues, thereby leading to liver cancer and death in some cases [9,10]. The toxin AFB1 primarily exists in agricultural products such as grain and oil [11]. Due to being present in animal feed, it also accumulates in the body of animals. It is subsequently metabolized into aflatoxin M1 (AFM1), which is then suspended in the milk of breastfeeding animals. The toxicity and carcinogenicity of AFM1 are similar to those that characterise AFB1 [12]. Because milk is a significant food source for human beings, especially infants and young children, it is necessary to have the appropriate systems and methods in place by which AFM1 levels can be monitored to allow for early warnings to indicate the presence of contamination and their associated risks to consumers [13].

At present, the commonly used pretreatment process for detecting AFM1 in food samples includes extraction, purification, and derivatization, among other steps. The techniques include liquid–liquid extraction, ultrasonic or microwave-assisted extraction, solid phase extraction, and immunoaffinity extraction [14,15]. Among these, the immunoaffinity extraction method has become a common method for detecting AFM1 due to its strong specificity. With this feature, this method has become the standard used by many governments and international organizations [16]. However, due to the specificity of affinity chromatography, it has several limitations: its degree of affinity is saturated to a certain extent, the stability of the affinity column possesses some deviation such that the method has a degree of uncertainty, and there is a high cost associated with the method. Most concerningly, the method leads to low recovery of AFM1, thereby affecting the accuracy of results [17]. Generally, at the trace level, the recovery for this method has been reported to be 70–80%, and in some cases, as low as 60% [18]. In the milk sample matrix, the primary two interfering substances that exist within it are protein and fat, which are also important sources of uncertainty in detecting small molecular substances in the sample [19]. Egg white is rich in amino acids, proteins, and water. In food processing, egg white is often used as a purification material to remove macroscopic impurities in food, especially those comprised of proteins and fats. It is also used in the routine detection of small molecular pollutants such as antibiotics and veterinary drugs, which are often enriched in egg white [20]. Studies have found that viscous polysaccharides (mannose, mucin, and albumin) that are present in viscous proteins (such as ovalbumin and albumin) are enriched in the small and medium molecules present in egg white [21,22,23,24]. Suppose egg white can be used as an effective matrix-purifying material. In that case, detecting trace levels of AFM1 in the complex milk matrix may be achieved, as well as developing a new method for “green environmental chemistry”.

This study aimed to develop a method to extract trace residues of AFM1 in milk using a hydrogel formed from concentrated egg white. UPLC-MS/MS was used for quantitative detection of AFM1 via simple protein salting out purification. The results were analyzed in parallel with the standard method of immunoaffinity clean-up. The findings show the novel method to be superior to the reference method in extraction recovery, sensitivity, repeatability, and operability. The new method of using egg white can therefore be used as a simple method for the routine surveillance of AFM1 in milk.

## 2. Results and Discussion

### 2.1. Preparation of the Blank Matrix Extract

In this experiment, UPLC-MS/MS was used to perform the quantitative analysis, and measurements obtained by this technique can be significantly affected by the food matrix analyzed. Therefore, a series of matrix-matched standard working curves were prepared to assess such matrix effects. Considering there may be AFM1 contamination within the matrix itself, before confirming the negative blank matrix samples using UPLC-Q-TOF, the blank matrix was further treated to ensure that no residual target analytes in the matrix could interfere with the optimization and evaluation of the method. The treatment involved the addition of 0.005 g sodium bicarbonate (NaHCO_3_) to each 5 g of milk to eliminate any trace levels of AFM1 that may have been present in the samples. An appropriate amount of formic acid was also added to neutralize NaHCO_3_. Verifications were carried out to ensure that the processing method did not affect the subsequent detection measurements. 

The method that was used is described as follows. Milk was supplemented with 1 μg/kg AFM1, and then 0.005 g of NaHCO_3_ was added to each 5 g of the spiked sample and mixed evenly. A volume of 0.5 mL of formic acid was then added for the neutralization step. The matrix effects of sample extraction solutions were compared between the above-pretreated samples and the negative sample that was only confirmed by UPLC-Q-TOF. The results of the quantitative analysis showed that the ion suppression effect of the two samples was within the range of 40 ± 4% (considered relatively consistent), indicating that the above pretreatment did not affect the results.

### 2.2. Selecting the Protein Precipitator

Egg white is an uneven, viscous transparent substance with a slightly yellow tinge. After simple low-speed centrifugation (less than 2000 r/min), it separates into two layers; relative to one another, the upper layer is thin while the lower layer is viscous, and these layers have a volume ratio of approximately 3:1. After mixing and centrifugation, this double-layer cannot be re-established. This phenomenon of irreversibility demonstrates that natural egg white possesses two layers of gelatinous substances formed due to the differential content of water and proteins rather than the distinct layers being characterized by distinct compounds. Therefore, the whole egg white was slowly shaken and mixed in this experiment. Various chemicals were tested for their ability to induce protein precipitation in exploring the treatment method to prepare egg white. These chemicals were methanol, acetonitrile, and inorganic salts. The results showed that the two organic solvents, methanol and acetonitrile, primarily acted by causing protein denaturation, thereby resulting in a relatively dense precipitate. Conversely, the protein precipitation produced by inorganic salts was characterized by a relatively more diffuse texture.

To further evaluate extraction performances, the following method was carried out. A standard solution containing AFM1 was added to egg white, and after the mixing and incubation steps were carried out, an AFM1-spiked sample containing a concentration of 1 μg/kg was obtained. Protein precipitation was carried out by denaturation and salting out, respectively, and the solution obtained after centrifugation was filtered to subsequently be analyzed by UPLC-MS/MS. The comparison of the results (Figure 1) showed that the recoveries after the salting out step all exceeded 90%. This showed that the protein in egg white exhibited no obvious adsorption to AFM1. The extraction effect of the salting out treatment was superior to that of protein denaturation (which obtained recoveries of less than 85%). Therefore, the ratio of the two salts (sodium chloride and ammonium acetate) used in the salting-out process was further optimized. The study found that after 10,000 r/min centrifugation, both a liquid and protein layer could be obtained. The precipitated protein produced by salting out with sodium chloride (NaCl) was distributed in the solution’s upper and lower layers. At the same time, that was obtained by ammonium acetate (NH_4_Ac) resided only in the lower layer. The textures of the mixtures produced by the two respective methods were also different—the one produced by NH_4_Ac was more diffuse than that by NaCl. The use of NH_4_Ac offered two advantages: less was used for the method (0.8 times that of NaCl), and the extraction recovery was superior. To avoid the potential loss of target components caused by any adsorption of dense materials, NH_4_Ac was selected as the main reagent for the salting out step. The above test also showed that the viscous state of egg white was the manifestation of the protein contained in the gelatinous fluid.

Similar steps were used to salt out the milk. The study found that the protein in the milk was located in the upper layer of the liquid after centrifugation. After mixing the milk and egg white, NH_4_Ac was added for salting out, followed by centrifugation, whereby the protein in the egg white aggregated in the upper layer of the liquid. This indicates that although the protein density of the egg white was higher than that of milk, the two types of proteins interacted and were mixed during the salting-out process and hence could be removed simultaneously.

### 2.3. Preparation of the Egg White Gel

Due to the greater proportion of egg white used in the above simple treatment process, and considering that the large amount of water contained in egg white increases the dilution ratio in the analysis process to affect the sensitivity of the method, it was necessary to perform further treatment steps to obtain a thick gel with lesser water content for its subsequent utilization to remove impurities in milk. Therefore, the methods for decreasing the water content of egg white were assessed. The egg white was concentrated by both slow stirring and nitrogen purging to obtain a gel, thereby reducing the water content from 88% to 10%. We also attempted to rehydrate commercially available egg white powder with water for reconstitution, but this method did not form an ideal hydrogel state. The reason for this result may be that the structural state of the protein had been changed or that denaturation was induced by dehydration during the production of the egg white powder, which then affected the quality of the rehydrated gel. Because milk contains water, the gel can be well dispersed after its addition to milk, and the dilution of the sample was effectively minimized.

### 2.4. Optimization of the Sample Pretreatment Method

To increase the protein removal efficiency, we added acid during the treatment process to encourage protein precipitation. Formic acid and acetic acid were selected for comparison to avoid mass spectrometry contamination. It was found that the protein states obtained by the two reagents were similar, though acetic acid was consumed in a larger quantity compared to formic acid. Therefore, formic acid was selected to be used as the auxiliary salting-out reagent to further promote protein removal.

Different proportions of milk and egg white were mixed. These mixtures formed an evenly turbid liquid. Formic acid and NH_4_Ac were then added to scavenge for proteins. After centrifugation at 10,000 r/min for 15 min, solutions containing two layers (precipitated protein and transparent liquid) were obtained. The results showed that the yellow tinge of egg white itself was lost after protein precipitation, thereby transforming the solution into a transparent, colorless liquid.

Subsequently, we optimized the ratio of gel to milk based on the matrix effect. These results are shown in Table 1. The preliminary tests showed that egg white could be used as a scavenger for milk. When the egg white gel to milk ratio was 5 g/10 g, an ideal extraction effect was obtained, and there was no obvious improvement in increasing the ratio further.

### 2.5. Comparison of the Clean-Up Effect

The clean-up effects resulting from the two methods used in this study were compared. Firstly, one solvent standard curve and two kinds of matrix matching standard curves were prepared. The results showed that the matrix matching standard curves prepared by egg white gel and immunoaffinity were linear in the range of 0.01 μg/kg to 30.0 μg/kg and 0.015 μg/kg to 30.0 μg/kg, respectively (R > 0.999). When the concentration exceeded 30.0 μg/kg, the trend appeared to show a saturation effect. The minimum concentration (LOQ) of the egg white gel method was 0.01 μg/kg, and the LOD was 0.001 μg/kg, while the LOQ and LOD of the immunoaffinity method were greater (0.015 μg/kg and 0.005 μg/kg, respectively). A comparison between the matrix effects showed that the test solutions obtained by the two pretreatment methods had a certain ion inhibition effect, whereby the ratio of the slope of the matrix matching standard curve to the solvent standard curve of the immunoaffinity method was 81%, thereby indicating that the inhibition effect was about 20%. The ratio of the egg white gel method as was determined by the same calculation method was 93%, and the corresponding inhibition effect was less than 10%, indicating that the egg white gel treatment yielded a superior degree of purification. The intercept of the three curves in descending order was that of the immunoaffinity treatment, egg white gel clean-up, and direct solvent preparation, while the corresponding proportion was approximately 5:2:1. This further illustrates the purification efficiency of egg white gel. The acidification and inorganic salt precipitation steps were also carried out without the addition of egg white gel, which then underwent centrifugation and filtration steps before the negative milk extract was tested by the MRM (multiple reaction monitoring) mode of mass spectrometry. The measured response of the non-target component was approximately 10^4^, which then decreased to 10^3^ after adding the egg white gel purification step. The measured response intensity of the sample obtained by the immunoaffinity column approach was higher than that obtained by the egg white gel method, which further confirmed the superior purification outcome of using egg white gel. Figure 2 shows the MRM chromatograms of the blank matrix processed by each of the three methods.

### 2.6. Influence of Acidification on the Analysis Results

Substances in milk that interfere with detecting AFM1 mainly include macromolecular proteins, fats, sugars, and other molecules. After optimizing the experimental conditions, this study investigated the removal efficiency of interfering substances as well as their interference effects in the co-elution of mass spectrometry-based detection methods, particularly regarding the proteins in milk. Other components, such as lipids, negatively affected the associated transfer dilution. Therefore, the interference caused by the protein content on the target component was further studied.

Comparing the two methods used in this study, immunoaffinity purification did not account for the influence of protein binding on the detection of AFM1 and, to a certain extent, detected AFM1 in its free, weakly bound and unencapsulated states. In addition to high-temperature heating, we found that increasing the acid content could effectively dissociate AFM1 in both the combined and cooperated states. Adding formic acid during the researched treatment method effectively released AFM1 while removing the protein component. Taking the detection results as an example, in the positive milk samples, the detection value for the immunoaffinity purification method was 0.91 ± 0.05 μg/kg (*n* = 5). At the same time, the corresponding recovery rate was 85%, and the detection value for the egg white gel purification method was 1.10 ± 0.02 μg/kg (*n* = 5), while the corresponding recovery rate was 93%. Several other laboratories were invited to test the samples using the standard method involving immunoaffinity purification to verify the accuracy of the test results and reduce the impact of human error and laboratory-specific factors on the results. Their results showed that the average detection value for the test results was 1.00 ± 0.25 μg/kg (*n* = 6), with a recovery rate between 70–85%.

There were clear differences between the results obtained by different laboratories, and it is speculated that the bound AFM1 in the positive samples affected these detection results. Additionally, the spiked recovery experiment results of parallel operations from each laboratory showed that material loss may have resulted from the immunoaffinity purification method. To further verify this, we used a quality control sample for comparison. The detection values of the two methods by either immunoaffinity or egg white gel purification were 2.57 ± 0.12 μg/kg and 3.09 ± 0.07 μg/kg, respectively (*n* = 5). Twelve laboratories were invited to collaborate on calibrating the quality control samples using the researched method. The detection values were 3.05 ± 0.28 μg/kg (*n* = 12). This result was not significantly different from the results obtained by our laboratory. The detection value obtained using the egg white gel purification method was higher than that of the immunoaffinity purification method. Compared with immunoaffinity purification, the matrix effect resulting from the egg white gel purification was smaller. Thus, it was determined that there was bound AFM1 present in the sample, and the addition of formic acid could effectively release it.

### 2.7. Method Validation

#### 2.7.1. Linear Range, LOD, and LOQ

The LOD and LOQ were taken as the peak response values of the recovered samples at 3× noise and 10× noise, respectively. The results were based on the actual identifiable signal. The corresponding linear equation, correlation coefficient, LOD, and LOQ are shown in Table 2. The analysis was repeated five times at each concentration point on the standard curve, and the slope RSD% was determined to be ≤2.3% in all cases, while the peak area deviation of the lowest point was ≤4.1% in all cases. The standard curves of the research method and the reference methods showed good linearity in the range of 0.01 to 30 μg/kg and 0.015 to 30 μg/kg, respectively (correlation coefficient R > 0.999). The LOD and LOQ of the egg white gel purification method were lower than those obtained by the immunoaffinity purification method. Figure 3A,B show the extraction ion chromatograms of the spiked milk samples with 0.001 μg/kg and 0.01 μg/kg of AFM1, respectively, following the egg white gel purification method. The results met the analysis requirements for the LOD (S/N = 3) and LOQ (S/N = 10). Figure 4A,B shows the corresponding chromatograms obtained by the immunoaffinity purification method of the AFM1-spiked milk samples at concentrations of 0.005 μg/kg and 0.015 μg/kg, respectively. This showed that, unlike the egg white gel method, the immunoaffinity method would need to increase the concentration in order to meet the analysis requirements of the LOD and LOQ.

#### 2.7.2. AFM1-Spiked Recovery and Reproducibility

An appropriate amount of AFM1 standard solution was added to blank milk samples at the LOQ, 10 × LOQ, 50 × LOQ, 100 × LOQ, and 200 × LOQ. The corresponding matrix matching standard curve was used for quantitative analysis. Table 3 shows that the recoveries of the 5 concentrations suspended in milk that were processed by the immunoaffinity purification method were within the range of 75.0–86.2%, with a RSD of 7.1–14.3% (*n* = 6). Recoveries obtained by the egg white gel purification method were within the range of 88.3–94.7% with a RSD of 6.1–11.0% (*n* = 6). These results met the requirements for routine sample analysis. The egg white gel purification method performed with better accuracy and repeatability due to its simpler method and higher purification.

#### 2.7.3. Stability and Daytime Precision

Sample stability and daytime precision were investigated. Immunoaffinity purification and egg white gel purification were both stable within 72 h. The RSD% of the five concentration levels obtained by the immunoaffinity purification method was between 7.5–11.2% (*n* = 5), and the daytime precision was between 10.3–14.6% (*n* = 5). Using the egg white gel purification method, the RSD% of the five concentration levels was between 6.6–13.1% (*n* = 5) and the RSD% of the daytime precision was between 7.8–10.5% (*n* = 5). In summation, both methods met the requirements for routine sample analysis.

### 2.8. Actual Sample Analysis

UPLC-MS/MS was used for the quantitative analysis of AFM1 in 30 milk samples obtained from the market using the above two pretreatment methods. Results showed that AFM1 was detected in 1 of the 30 samples. The detected values obtained from the egg white gel method and the immunoaffinity purification method were 1.10 ± 0.02 μg/kg (*n* = 5) and 0.91 ± 0.05 μg/kg (*n* = 5), respectively, both of which exceeded the maximum residue limit (0.5 μg/kg) set for milk and dairy products by the EU and China. The detection result obtained from using the egg white gel purification method was slightly higher than that obtained by the immunoaffinity purification method.

### 2.9. Inter-Laboratory Validation

According to the Chinese food composition table compiled by the Chinese Centers for Disease Control and Prevention [25], the nutritional value of eggs is roughly the same: the egg white contains 87% water, with the remaining 13% as solids. 90% of the solids are proteins, including 75% ovalbumin, 15% ovalbumin, 7% ovalbumin, and 3% albumin. It was found that the structure of sticky polysaccharides in these sticky proteins played a role in the enrichment of small and medium-sized molecules in egg white. Therefore, it is speculated that the composition of different batches of egg white is essentially the same and hence not impactful enough to affect the experimental results.

However, to further investigate the applicability, accuracy, and repeatability of the method for other researchers, we invited 12 laboratories based in different provinces of China to purchase eggs from local markets as raw materials and follow the egg white gel preparation method that was developed for this paper, then to analyze the same one quality control sample using the researched method. The result was 3.05 ± 0.28 µg/kg (*n* = 12), and the relative standard deviation was less than 10%. There was no significant difference between the result of our laboratory (3.09 ± 0.07 μg/kg).

### 2.10. Recommendations for the Immunoaffinity Purification Method

Immunoaffinity chromatography is widely used to detect AFM1 in milk [26]. In addition to the reference method in this study, such as European Standardization Committee EN ISO 14501 [27] and so on. In this study, quality control samples for milk were processed using the two methods and then tested. The differences in their obtained results revealed that there was adsorption or binding interactions between AFM1 and milk protein. According to the investigation of different AFM1 spiking concentrations, this phenomenon was more pronounced in the sample of a higher concentration level (above 1 μg/kg). According to these results, the initial calculation of the binding rate showed that the apparent binding rate fluctuated in the range of 5–10%, suggesting that the reference method used for the immunoaffinity chromatography purification method may need to further consider the completeness of the sample extraction steps.

In this experiment, only milk samples were fully researched. Other food samples contain greater degrees of macro- and microphysical interferences, so the immune affinity purification method is still the recommended method for determining AFM1 in other foods. Suppose those who produce the standards could improve the extraction steps of the immunoaffinity method on the existing basis and effectively release the AFM1 that exists as bound to protein. In that case, the method’s accuracy will be further improved.

### 2.11. Further Improvement of the Research Method 

We found that currently available market-sold egg white powder is redissolved in water as an extract, which did not meet the requirements for extraction because it could not form a suitable hydrogel. Therefore, it is necessary to use fresh eggs to prepare egg white gel. In this process, attention should be paid to avoiding the introduction of yolk so as not to affect the repeatability of the sample purification process.

The egg white gel was prepared in the laboratory for analysis and the verification of the methodology of AFM1 in milk. Next, we will further study the simultaneous detection of a variety of mycotoxins across multiple species of animal-derived samples such as muscle and viscera. On this basis, we will also optimize the process parameters after scaling up the production in order to realize the application of this environmentally friendly material in a wider range of fields [14,28].

## 3. Materials and Methods

### 3.1. Chemical Reagents and Samples

The AFM1 Immunoaffinity column (Romer Corporation, Lawrenceburg, TN, USA), AFM1 standard (99.9% purity, Romer Corporation, Lawrenceburg, TN, USA); Formic acid, acetic acid (chromatography grade, Sigma Company, New York, NY, USA); methanol, acetonitrile (chromatography grade, Thermo Fisher Technology Company of the United States, Waltham, MA, USA); NaHCO_3_, NaCl, NH_4_Ac (analytical grade, China Sinopharm Chemical Reagents Co., LTD., Shanghai, China). Ultrapure water was obtained using a Milli-Q water polishing system.

The 30 commercial fresh milk samples were purchased from a local market in Beijing, China, which included a milk sample positive for AFM1 (The reference value was assigned by inviting six laboratories to conduct collaborative testing, and the median value was 1.00 ± 0.25 μg/kg), and a milk quality control sample (obtained by feeding contaminated feed to cows in cooperation with Romer Corporation of the United States. The reference value was assigned by inviting 12 laboratories to conduct collaborative testing, and the median value was 3.05 ± 0.28 μg/kg).

### 3.2. Analytical Instrumentation

UPLC-MS/MS was performed using the Xevo TQ-S in conjunction with: the MassLynx mass spectrometry software (Waters Corporation, Milford, MA, USA); UPLC-Q-TOF 6540 (Agilent Corporation, Santa Clara, CA, USA); nitrogen blowing instrument (Anpu Corp. Ltd., Shanghai, China); CF16RXII centrifuge (HITACHI Corporation, Tokyo, Japan); chromatographic conditions: chromatographic column: Waters Aquity UPLC HSS T3, 2.1 × 100 mm, 1.8 µm; mobile phase: A, 0.1% (*v*/*v*) formic acid aqueous solution; B, acetonitrile. Gradient elution: 0.0–0.5 min, 27% B; 4.2 min, 40% B; 5.0–5.7 min, 100% B; 6.0 min, 27% B. The flow rate was 0.3 mL/min; column temperature: 40 °C; injection volume 10 μL; analysis time: 6 min.

The mass spectrometry conditions were as follows: the ionization mode used the electrospray positive ion mode (ESI+); the scanning mode used multi-ion reaction monitoring (MRM); the capillary voltage was 3.2 kV; the taper hole voltage was 40 V; the ion source temperature was 150 °C; the deflection voltage was 70 V; the cone hole gas flow was 150 L/h; the dissolvent temperature was 500 °C; the desolvent gas flow rate was 550 L/h; the ion condition was 329.1/273.1 (quantitative ion pair, collision energy 20 eV), 329.1/259.1 (qualitative ion pair, collision energy 25 eV).

### 3.3. Sample Pretreatment

#### 3.3.1. Reference Method: Immunoaffinity

According to GB 5009.24 [29], the basic steps were to weigh 4 g milk in a 50 mL centrifuge tube, then 10 mL methanol was added and extracted by vortexing for 3 min. The milk was centrifuged at 6000 r/min at 4 °C for 10 min. The supernatant was transferred to a beaker, diluted with 40 mL water, injected into the immunoaffinity column, washed with 10 mL water, drained, and then eluted with 4 mL acetonitrile. The eluent was blown to near-dry with nitrogen at 50 °C and then redissolved to a volume of 1.0 mL with the initial mobile phase. It was then filtered by a 0.22 µm microporous membrane for subsequent testing.

#### 3.3.2. Research Method: Egg White Gel

Ten eggs were separated to obtain their egg whites (totaling approximately 260 mL in volume). These were then placed in a beaker and mixed evenly. Approximately 50 mL of the egg white mixture was then transferred to another beaker with a magnetic rotor and stirrer for slow stirring and heating at 37 ± 3 °C. Following 5 h of nitrogen purging, the egg white was concentrated at approximately 5 mL. The thick portion of egg white at the top was removed and set aside after centrifuging at 8000 r/min for 20 min. This was repeated to obtain approximately 10 g (20 mL) of thick egg white gel material. Then, 10 g of pre-mixed milk sample was placed in a 50 mL centrifuge tube, and 5 g of the egg white concentrate was added to this. After evenly mixing, 2 mL of formic acid solution and 4 g of NH_4_Ac were added. After centrifuging the mixture at 8000 r/min for 20 min, it was then filtered by a 0.22 µm microporous membrane for subsequent testing.

### 3.4. Preparation of the Blank Matrix Extract

Milk samples that were unknown to contain AFM1 were pretreated by adding 0.005 g NaHCO_3_ to each 5 g of sample, then mixed evenly, added with 0.5 mL of formic acid, and then thoroughly mixed once more. Two differentially prepared blank matrix extraction solutions were produced according to Section 3.3.1 and Section 3.3.2, respectively. No corresponding chromatographic peak appeared at the same retention time as the standard AFM1 substance, and this was confirmed by high-resolution mass spectrometry, thus indicating no detectable AFM1 in the sample.

## 4. Conclusions

In this study, we developed, performed, and assessed a novel method for complex matrix sample purification using natural biological materials combined with a UPLC-MS/MS detection strategy to determine the content of AFM1 residue in milk. Egg whites were concentrated and prepared to form gels, which were then mixed with milk samples in appropriate proportions, followed by the addition of formic acid and NH_4_Ac to remove proteins and other substrates that interfered with the determination of the target analytes. Compared with the immunoaffinity technique used as the reference method, the novel egg white method developed in this study showed advantages in extraction recovery, sensitivity, repeatability, and operability. The LOQ was reduced from 0.005 µg/kg to 0.001 μg/kg, while the LOD was reduced from 0.015 µg/kg to 0.01 µg/kg, and the recovery was increased from the range of 75.0–86.2% to 88.3–94.7%.

Additionally, by significantly reducing the operation time and requirements for operation, the new method better fulfilled the desirable criterion of being based on “green analytical chemistry”. This study provided a simple, sensitive, and accurate method for determining AFM1 in milk and provided a reference for further improving relevant international standards. It also provides more practical prospects for applying natural substances such as egg white in measuring substances within complex matrix food samples.

## Figures and Tables

**Figure 1 molecules-27-05039-f001:**
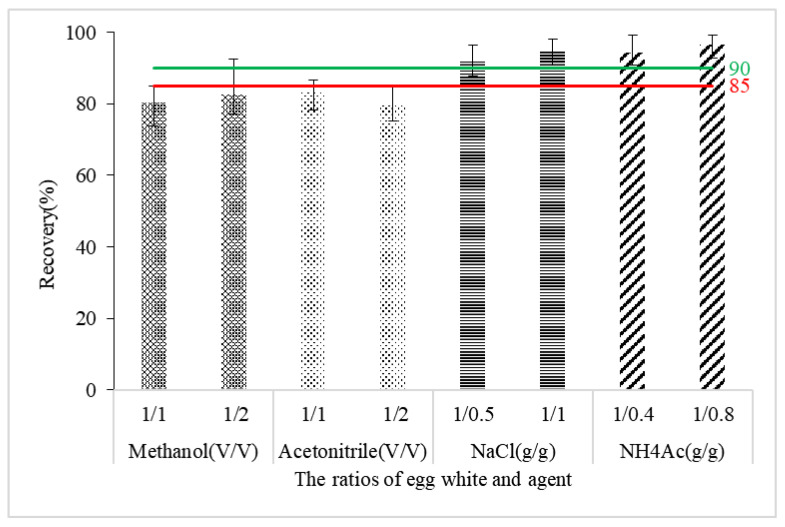
Comparison of different types and proportions of protein precipitants (*n* = 5).

**Figure 2 molecules-27-05039-f002:**
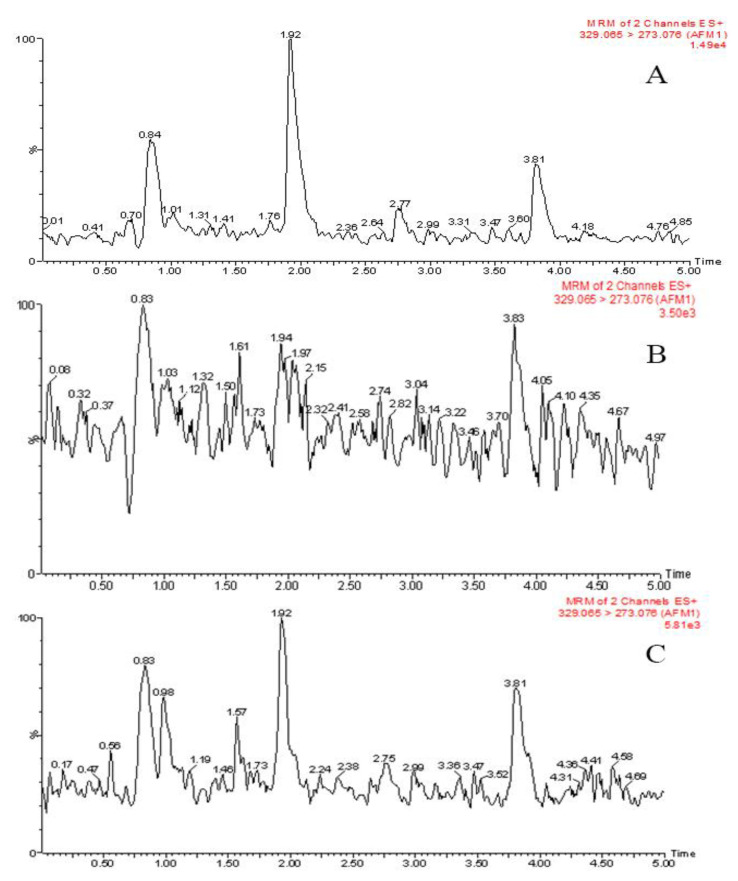
Extracted ion chromatograms of the blank matrix were treated in three different ways as described in Section 2.5 ((**A**): acidification and salting out, (**B**): egg white gel, (**C**): immunoaffinity column).

**Figure 3 molecules-27-05039-f003:**
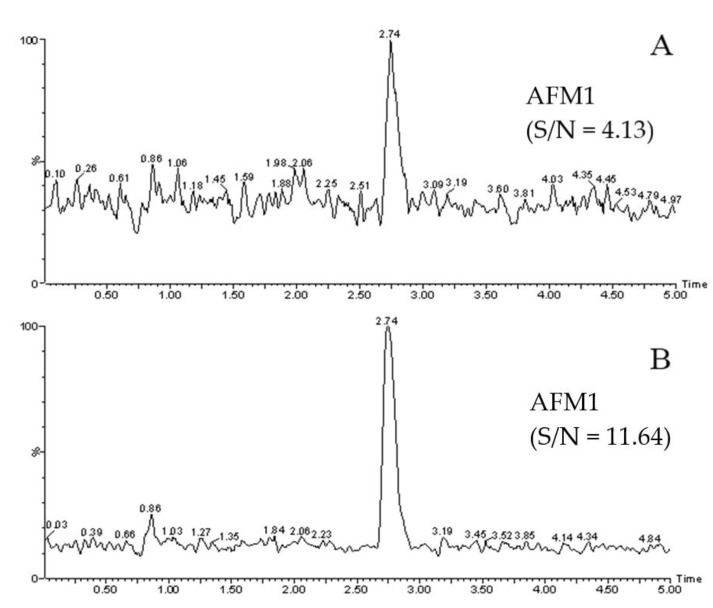
Extracted ion chromatograms of AFM1-spiked milk by the egg white gel method clean-up ((**A**):0.001 μg/kg, (**B**):0.01 μg/kg).

**Figure 4 molecules-27-05039-f004:**
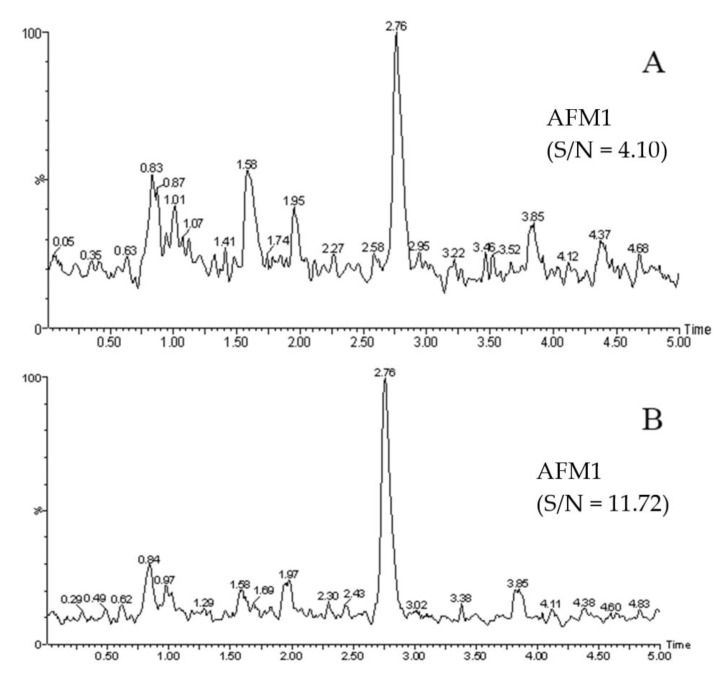
Extracted ion chromatograms of AFM1-spiked milk by the immunoaffinity method clean-up ((**A**): 0.005 μg/kg, (**B**): 0.015 μg/kg).

**Table 1 molecules-27-05039-t001:** Comparison of different types and proportions of protein precipitants (*n* = 5).

Gel Added into Milk (g/10 g)	Matrix Effect for 1 μg/kg AFM1 Spiked Milk (%)	Average Recoveries (%)	RSD (%)
1	85.3	87.3%	3.7%
2	88.6	89.0%	4.1%
5	92.1	95.4%	3.0%
10	91.4	96.1%	3.4%

**Table 2 molecules-27-05039-t002:** Regression equation, R, linear range, LOD, and LOQ for LC-MS/MS analysis of AFM1 (*n* = 5).

Method	Linear Range (μg/kg)	Regression Equation	Correlation Coefficient (R)	LOD (μg/kg)/ RSD(%)	LOQ (μg/kg)/ RSD(%)
Immunoaffinity	0.015~30	y = 512,638x + 2614	0.9994	0.005/4.1	0.015/3.0
egg white gel	0.01~30	y = 588,584x + 1027	0.9992	0.001/2.8	0.01/3.3

**Table 3 molecules-27-05039-t003:** Average recovery and precision of AFM1 in milk (*n* = 6).

Method	Spiked Level (μg/kg)	Recovery (%)	RSD (%)
Immunoaffinity	0.015	82.6	14.3
	0.15	86.2	12.5
	0.75	84.1	7.1
	1.5	77.5	8.6
	3	75.0	7.4
Egg white gel	0.01	88.3	10.2
	0.1	92.2	11.0
	0.5	90.9	9.7
	1	94.7	6.1
	2	91.4	8.3

## Data Availability

All the data are included in this manuscript.

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
