# Peer review of "A Strategy for Sample Preparation: Using Egg White Gel to Promote the Determination of Aflatoxin M1 Content in Milk Samples"

_molecules, 2022, doi:10.3390/molecules27155039_

Round 1

Reviewer 1 Report

The article proposes a great alternative to the conventional method approved for the analysis of AFM1 in milk, in the view of green chemistry and without losing the analytical performance required for a correct characterization.

The experimental workflow is well structured and the results are clearly presented and supported by figures and tables.

I suggest a general English revision to improve the article's soundness and increase the references and the discussion of the results in comparison with other scientific articles in the same field.

I propose a minor revision, the paper can be considered valid for publication after small changes. 

Reviewer 2 Report

The paper titled “A Strategy of Sample Preparation Using Egg White Gel to Promote the Determination of Aflatoxin M1 Content in Milk Samples” investigated the possibility to employ hydrogel formed from concentrated egg white as extraction step for AFM1 quantification method. The topic is novel and scientifically relevant.

The manuscript is well-structured and clearly written. Figures and tables are clear and well-presented. Cited references are relevant and recent. Nevertheless, the manuscript needs to be reviewed for some minor technical corrections.

Title is suitable and illustrates the aim of the research.

Introduction is concise and informative, indicating the background and the aim if the research.

Results are presented in clear and understandable manner, and comprehensively discussed.

Methodology is described in detail, enabling the reader to replicate the research. The method validation methodology was appropriately selected and adequately presented.

Conclusions are derived from the results and well-presented. However, it would be useful if the authors indicated any weaknesses or limitations that employment egg white gel as extraction step may have.

Reviewer 3 Report

General

I question the use of a natural product for this application.  The results of aflatoxin analysis, especially in a commercial or regulatory context has the potential for great economic or health impact.  Being accurate is obviously important, but a high degree of repeatability is similarly important.  The present study address the accuracy and repeatability, within their lab, but how repeatable would this be by other researchers?  Even if they followed the same methods, would the eggs / egg white / egg white gel be the same if they were sourced from another market?  Some refined natural products (say, corn starch) are fairly standardized, but I don’t know how much variation there is eggs.  From the Materials & Methods section here, I can’t see if you tested it beyond a single batch of 10 eggs, and I’m pretty sure you didn’t test it with eggs from another location.  

Line 17:  “greener” is super vague.  It means different things to different audiences.  Does it mean less toxic?  Less energy?  Generates less waste volume?  Just say what you mean.

Line 22:  When you say that the method has advantages, compared to what?  If it is ‘better’ what is it better than?

Line 39:  Again, “greener” is too vague.  

Line 42:  Aflatoxin is not “considered to be a toxic substance”.  It IS toxic.  This is objective fact, not an opinion.

Line 46:  Not really a matter of “migrating”.  Maybe “accumulates in” or “contaminates meat and dairy products” 

Line 68:  Delete “Being a natural biological material” from the sentence.  Lots of natural biological materials are NOT rich in amino acids, proteins and water.

Line 95:  Why a “certain amount”?  Just say how much.  Cryptic.

Lind 192:  “Bending effect”?  Maybe “plateauing”  or “saturation”?

Section 2.6:  Generally this section is well-written and well thought-out.  Possibly re-write the sentences from 235-237 – was this with a Romer program?  What was the nature of this inter-lab validation?
